# Enhancing rice growth and yield with weed endophytic bacteria *Alcaligenes faecalis* and *Metabacillus indicus* under reduced chemical fertilization

**Kaniz Fatema**[1], **Nur Uddin Mahmud**[1], **Dipali Rani Gupta**[1], **Md. Nurealam Siddiqui**[2], **Tahsin Islam Sakif**[3], **Aniruddha Sarker**[4], **Andrew G. Sharpe**[5], **Tofazzal Islam**[1]*

**1** Institute of Biotechnology and Genetic Engineering (IBGE), Bangabandhu Sheikh Mujibur Rahman Agricultural University, Gazipur, Bangladesh, **2** Department of Biochemistry and Molecular Biology, Bangabandhu Sheikh Mujibur Rahman Agricultural University, Gazipur, Bangladesh, **3** Keck Graduate Institute, Claremont, California, United States of America, **4** Residual Chemical Assessment Division, National Institute of Agricultural Sciences, Rural Development Administration, Jeollabuk-do, Republic of Korea, **5** Global Institute for Food Security, University of Saskatchewan, Saskatoon, Saskatchewan, Canada

* tofazzalsilam@bsmrau.edu.bd

## Abstract

Endophytic bacteria, recognized as eco-friendly biofertilizers, have demonstrated the potential to enhance crop growth and yield. While the plant growth-promoting effects of endophytic bacteria have been extensively studied, the impact of weed endophytes remains less explored. In this study, we aimed to isolate endophytic bacteria from native weeds and assess their plant growth-promoting abilities in rice under varying chemical fertilization. The evaluation encompassed measurements of mineral phosphate and potash solubilization, as well as indole-3-acetic acid (IAA) production activity by the selected isolates. Two promising strains, tentatively identified as *Alcaligenes faecalis* (BTCP01) from *Eleusine indica* (Goose grass) and *Metabacillus indicus* (BTDR03) from *Cynodon dactylon* (Bermuda grass) based on 16S rRNA gene phylogeny, exhibited noteworthy phosphate and potassium solubilization activity, respectively. BTCP01 demonstrated superior phosphate solubilizing activity, while BTDR03 exhibited the highest potassium (K) solubilizing activity. Both isolates synthesized IAA in the presence of L-tryptophan, with the detection of *nifH* and *ipdC* genes in their genomes. Application of isolates BTCP01 and BTDR03 through root dipping and spraying at the flowering stage significantly enhanced the agronomic performance of rice variety CV. BRRI dhan29. Notably, combining both strains with 50% of recommended N, P, and K fertilizer doses led to a substantial increase in rice grain yields compared to control plants receiving 100% of recommended doses. Taken together, our results indicate that weed endophytic bacterial strains BTCP01 and BTDR03 hold promise as biofertilizers, potentially reducing the dependency on chemical fertilizers by up to 50%, thereby fostering sustainable rice production.

**Data Availability Statement:** All relevant data are within the manuscript and its Supporting information files.

**Funding:** The author(s) received no specific funding for this work.

**Competing interests:** The authors declare no competing interests.

**Abbreviations:** IAA, Indole-3-acetic acid; KSB, potassium solubilizing bacteria; KSI, potassium solubilizing index; NB, Nutrient broth; PGPB, plant growth-promoting bacteria; PSB, phosphate solubilizing bacteria; PSI, phosphate solubilizing index.

# 1. Introduction

Rice (*Oryza sativa* L.), a staple for nearly half the global population and the third-largest cereal crop worldwide, holds paramount importance in sustaining human diets [1]. In 2020, the USDA reported global rice production at 503.17 million metric tons, utilizing 11% of cropland [2]. Bangladesh, ranking third in global rice production, dedicates around 78% of its arable land to rice cultivation, projecting an output of 38.4 million tons [3]. For developing countries, including Bangladesh, rice contributes significantly to daily caloric intake, providing 27% of dietary energy, 20% of dietary protein, and 3% of dietary fat [4]. However, the heavy reliance on agrochemicals, such as urea, triple super phosphate (TSP), and muriate of potash (MoP) for the higher yield of rice, poses environmental threats, prompting the exploration of sustainable alternatives [5]. Furthermore, the natural mineral sources for the production of these three major fertilizers required for rice production are finite and depleting day by day.

This study addresses the urgent need for low-cost technologies to enhance crop productivity while mitigating the environmental impact of chemical fertilizers. While various strategies exist, leveraging beneficial microbes offers an economical and viable solution [6–8]. Notably, plant growth-promoting bacteria (PGPB) or plant probiotic bacteria have emerged as promising contributors to enhanced productivity, particularly in rice cultivation [9]. A large body of literature indicates that plant-associated bacteria serve as biofertilizers and biostimulants, offering natural and renewable resources to reduce the use of hazardous synthetic chemicals in rice production [6–9]. The mechanisms of the plant probiotic bacteria include fixation of atmospheric nitrogen, solubilization of plant essential nutrient elements in soils, production of phytohormones and various metabolites and regulation of gene expression in the host plants. Some of these plant probiotic bacteria belonging to the genera of *Bacillus*, *Rhizobium*, *Pseudomonas*, *Enterobacter*, *Paraburkholderia*, *Delftia* etc. have been proven as biofertilizers and/or biopesticides in production of many crops including rice [6–17].

Despite the valuable role of plant endophytic bacteria in plant growth, their potential, especially weed endophytes, remains poorly underexplored [9–29]. The hypothesis of this study was weed endophytes from rice field can enhance growth and yield of rice under low fertilization conditions. This research aims to isolate and characterize endophytic bacteria from rice-associated weeds, assess their impact on rice growth and yield, identify the bacteria through 16S rRNA gene sequencing, and elucidate their growth-promoting roles by detecting genes involved in nitrogen fixation, phosphorus and potassium solubilization, and IAA production. While weed endophytes are often overlooked, their adaptation to diverse conditions makes them potential reservoirs of beneficial bacteria with unique capabilities, offering novel insights for sustainable agriculture.

# 2. Materials and methods

## 2.1. Experimental sites

The native weed samples were collected to isolate bacteria from the field laboratory at Bangabandhu Sheikh Mujibur Rahman Agricultural University (BSMRAU), located in Gazipur, Bangladesh (24.09˚ N and 90.25˚ E).

## 2.2. Collection of plant materials and isolation of bacteria

Root and shoot samples were collected from various weed species, including Ulu (Cogon grass: *Imperata cylindrica*), Chapra (Goose grass: *Eleusine indica*), Kashful (wild sugarcane: *Saccharum spontaneum*), Mutha (Nutsedge: *Cyperus rotundus*), Durba (Bermuda grass: *Cynodon dactylon*), Anguli Ghash (Scrab grass: *Digitaria sanguinalis*), Khude sama (Jungle Rice:

*Echinochloa colonum*), Arail (Swamp rice grass: *Leersia hexandra* Sw.), Boro shama (Barnyard Grass: *Echinochloa cruss-galli*), Kakpaya (Crow foot grass: *Dactyloctenium aegyptium*). These samples were collected at the vegetative stage from experimental sites where the weeds naturally grew, facilitating bacterial isolation. Additionally, seeds of the rice variety CV. BRRI dhan29 were procured from the Bangladesh Rice Research Institute (BRRI) for use in pot experiments to examine the isolated endophytic bacterial effect by assessing multiple parameters.

To prepare the root and shoot samples for isolation, thorough washing with distilled water, followed by a 5-minute rinse with 70% ethanol, was conducted. Subsequently, the samples were sterilized with 1% NaOCl for 1 minute, followed by a thorough rinse with sterile distilled water. The tissue was then further rinsed for 1 minute in 100% ethanol, followed by five washes with sterile distilled water.

The processed samples were crushed in a sterilized mortar and pestle, diluted with sterile distilled water (SDW) up to a $1 \times 10^{-6}$ dilution. A 100 μl aliquot of each dilution was evenly spread on Petri dishes containing nutrient agar medium and incubated for 2 days at 25˚C [6]. Colonies with distinct appearances were then transferred to new nutrient agar medium plates for purification. The purified isolates (single colonies) were preserved in a 20% glycerol solution at -20˚C.

## 2.3. Seedling assay

The bacterial strains were initially cultured in 250 mL conical flasks containing 200 mL of NB (Nutrient Broth) medium, placed on an orbital shaker at 120 rpm, and incubated for 72 hours at 27˚C. Subsequently, the resulting broth underwent centrifugation at 15,000 rpm for 1 minute at 4˚C, and the bacterial cells were collected and washed twice with sterilized distilled water (SDW). The bacterial pellets were then resuspended in 0.6 mL of SDW, vortexed for 45 seconds, and prepared for seed treatment.

For seed treatment, 1 gram of surface-sterilized rice seeds (CV. BRRI dhan29) was immersed in the bacterial suspension, dried overnight at room temperature, and arranged on a Petri dish with water-soaked sterilized filter paper. Following seed germination, the seedlings were allowed to grow for two weeks, receiving alternate-day watering. Germination percentages were calculated at two days after inoculation (DAI). After 15 DAI, the impact of plant probiotic bacteria on rice seedling growth was evaluated, recording parameters such as germination rate, shoot length (cm), root length (cm), shoot fresh weight (g), and root fresh weight (g).

## 2.4. Biochemical characterization of isolated bacteria

For the biochemical characterization of the isolated bacteria, the gram reaction was determined according to the method outlined by [30]. Various biochemical tests were conducted to characterize the isolated bacteria, following the criteria outlined by [31]. The assessment of KOH solubility involved mixing bacterial isolates with a 3% KOH solution on a clean slide for 1 minute, and the observation of a thread-like mass. Catalase and oxidase tests were performed following the procedures described by [32, 33].

## 2.5. DNA extraction, 16S rRNA gene amplification and phylogenetic analysis of isolated bacteria

The bacterial DNA extraction utilized the lysozyme-SDS-phenol-chloroform method with phenol-chloroform-isoamyl alcohol (25:24:1), followed by precipitation with isopropanol, following the procedure outlined by [34]. Subsequently, the extracted DNA underwent treatment

with DNase-free RNase (Sigma Chemical Co., St. Louis, MO, USA) at a final concentration of 0.2 mg/ml, incubated at 37°C for 15 minutes. Amplification of the 16S rRNA gene was achieved using a universal primer (27F, 5'AGAGTTTGATCCTGGCTCAG3'; 1492R, 5'GGTTACCTGTTACGACTT3') [35], and the reaction was carried out in a thermocycler (Mastercycler® Gradient, Eppendorf, Hamburg, Germany) following established guidelines.

The amplified products underwent purification using Quick PCR purification columns (Promega, Madison, WI, USA) and were subsequently sequenced with the Big Dye Terminator Cycle Sequencing Ready Reaction Kit on an Applied Biosystems analyzer (Applied Biosystems, Forster City, CA, USA). Sequences were compared to the NCBI GenBank database (http://www.ncbi.nlm.nih.gov) through a BLASTN search. For phylogenetic analysis, reference sequences were retrieved, and multiple sequence alignment was conducted using the CLUSTALW program in BioEdit version 7.2.3 [36], with manual editing of gaps. The construction of a phylogenetic tree employed the neighbor-joining method (NJ) [37] in the MEGA software package version MEGA7 [38]. Pair-wise evolutionary distances were calculated using the Maximum Composite Likelihood method [35], and confidence values, based on sequence grouping, were obtained through bootstrap analysis with 1000 replicates [39, 40].

## 2.6. Design of primers for amplification of *nifH* and *ipdC* genes

The *ipdC* gene is one of the key determinants that regulate bacterial indole acetic acid production through IPyA pathway and inactivation of this pathway results reduction of indole acetic acid (IAA) production up to 90%. Biological nitrogen fixation by bacteria is an important property contributing to plant growth. *nifH* gene encodes for nitrogenase enzyme catalyzes biological dinitrogen reduction to ammonium. Primers were meticulously crafted through homology searches for a specific gene *(nifH* and *ipdC)* within *Alkaligenes* spp. and *Metabacillus* spp., as documented in the NCBI GenBank. The primer pairs were designed based on the region exhibiting homology across these genera (Table S1 in S1 File).

## 2.7. Bioassays for plant growth promoting traits

**2.7.1. Determination of IAA production.** The determination of indole-3-acetic acid (IAA) production by two bacterial isolates followed the original protocol proposed by [41], with minor adaptations. In brief, isolated colonies were inoculated into 50 ml of sterile Jensen broth (comprising 20 g/l sucrose, 1 g/l K2HPO$_4$, 0.5 g/l MgSO$_4$ • 7H2O, 0.5 g/l NaCl, 0.1 g/l FeSO$_4$, 0.005 g/l NaMoO$_4$, and 2 g/l CaCO$_3$) [42]. The medium also contained 1 ml of 0.2% L-tryptophan. The cultures were incubated at (25 ± 2) °C for 72 hours with continuous shaking (100 rpm), alongside an uninoculated medium serving as a control. Following incubation, the cultures were centrifuged for 10 minutes at 12,000 rpm, and 1 ml of the clear supernatant was mixed with 2 ml of Salkowski reagent (comprising 50 ml of 35% perchloric acid and 1 ml of 0.05 mol/L FeCl3 solution). The mixture was then incubated in the dark at room temperature for 30 minutes. The change in color from visible light pink to dark pink indicated IAA production, and the absorbance at 530 nm was measured using a spectrophotometer. The IAA content was calculated using an authentic IAA standard curve.

**2.7.2. Screening for inorganic phosphate solubilization by isolated bacteria on agar assay.** All bacterial isolates underwent testing for mineral phosphate solubilization activity, employing the National Botanical Research Institute's phosphate (NBRIP) growth medium supplemented with 1.5% Bacto-agar (Difco Laboratories, Detroit, MI, USA) [43]. Triplicate inoculations of each bacterial isolate were carried out on NBRIP agar medium and incubated for 72 hours at (25 ± 2) °C. The capacity of the bacteria to solubilize insoluble tricalcium phosphate (TCP) was evaluated using the phosphate solubilization index (PSI) = A / B, where A

represents the total diameter (colony + halo zone), and B is the diameter of the colony [44]. The quantification of solubilized phosphorus (P) was determined by subtracting the available P in the inoculated sample from the corresponding uninoculated control [45].

### 2.7.3. Screening for mineral potash solubilization by isolated bacteria on plate assay.

The screening for mineral potassium solubilizing activity in all isolated bacteria was conducted using modified Aleksandrov media [46], incorporating insoluble potassium minerals. Each bacterial isolate was individually inoculated in a petri dish and incubated at 28°C for 7 days post-inoculation. The isolates were cultured in modified Aleksandrov media containing waste biotite at a concentration of 3g/l. The potassium solubilizing bacteria (KSB) were assessed based on the characteristics of their halo zones. After incubation, the measurements of the halo zone and colony diameter were recorded. The potassium solubilization capacity of the isolates was determined using the potassium solubilizing index (KSI) = A / B, where A represents the total diameter (colony + halo zone), and B is the colony diameter [44]. The quantification of solubilized potassium (K) was calculated by subtracting the available K in the inoculated sample from the corresponding non-inoculated control [45].

### 2.7.4. Assessment of growth and yield performances of rice grown in nutrient-deficit soil.

To assess the plant growth promotion capabilities of the two most effective probiotic bacteria, BTCP01 and BTDR03, a pot experiment was conducted using rice seeds (CV. BRRI dhan29) from November 2016 to May 2017.

The experimental soil, with a slightly acidic pH of 6.41 and clayey texture up to 50 cm depth, contained 0.08% total nitrogen (N), 9 mg/kg available phosphorus (P), 5.7 mg/kg soil-exchangeable potassium (K), and 1.55% organic matter. Meteorological data, including air and soil temperatures at a depth of 30 cm, and rainfall were obtained from the weather archive of the Department of Agricultural Engineering, BSMRAU. Throughout the crop's growing season, the maximum air temperature ranged from 23.5°C to 36.5°C, while the minimum air temperature ranged from 9°C to 27°C (Fig. S1 in S1 File).

Chemical fertilizers (2.10 g urea, 0.86 g gypsum, and 0.46 g zinc sulfate per 10 kg of soil) were applied based on the Fertilizer Recommendation Guide (FRG) for rice seed CV. BRRI dhan29. Triple superphosphate (TSP) and muriate of potash (MoP) were applied as a basal dose, with urea administered in three equal doses as top dressing at specific growth stages. Cultural practices, including weeding and irrigation, were performed as needed.

Forty-five-day-old seedlings with 3/4 leaves were transplanted into pots (20 cm × 20 cm × 30 cm), with one seedling per pot were cultivated separately in 1000 ml nutrient broth in conical flasks on an orbital shaker for 72 hours. A single colony from the regularly maintained bacterial culture plate was used for inoculation. The flask was then placed on a shaking incubator set at 120 rpm and 25°C for 72 hours to facilitate bacterial growth in the nutrient broth. The resulting bacterial suspension was serially diluted to achieve a concentration of $1 \times 10^9$ CFU/mL using sterile distilled water. Prior to transplantation, the seedlings were uprooted, cleaned, and their roots were immersed overnight in the bacterial suspension to enhance root colonization. During the tillering and flowering stages, the roots of the seedlings were again immersed in the bacterial suspension ($1 \times 10^9$ CFU/mL) overnight, and freshly harvested bacteria were sprayed onto rice plants [9].

The experiment was set in a completely randomized design with three replications which included untreated control i) and treatments with BTCP01 (ii) and BTDR03 (iv) using 0%, 50%, and 100% doses of recommended N, P, and K fertilizers. Essential plant growth parameters were recorded, including root and shoot length, root and shoot dry weight, SPAD value of the flag leaf at panicle initiation, number of tillers and effective tillers per plant, and total grain weight per pot.

## 2.8. Statistical analysis

The data obtained from seedling assays, the P and K solubilizing study and the pot experiments underwent analysis of variance using SPSS (version 17.0) and Statistix (version 10.1). Statistical differences among mean values were determined using the least significant difference (LSD) test at a 5% probability level. The presented data represent mean values ± standard error.

## 2.9. Additional information

This study requires no permits from any authority as the endophytes were isolated from the weeds of cultivated crop field and the laboratory of the Institute of Biotechnology and Genetic Engineering (IBGE) has standard facilities for such kind of research.

# 3. Results

## 3.1. Isolation, biochemical and molecular characterization of weed endophytic bacteria

A total of 45 bacteria, exhibiting diverse shapes and colors of colonies on nutrient agar medium plates, were isolated from surface-sterilized shoots and roots of the collected weeds. These isolates were subsequently purified through repeated streak cultures on NBA medium (Fig. S2 in S1 File). Their impact on seed germination and seedling growth of rice remarkably varied, with some isolates demonstrating inhibitory effects on rice seed germination, as illustrated in Figure S2 in S1 File. Following comprehensive screening, two strains, namely BTCP01 and BTDR03, were chosen based on their superior effects on seed germination rate, shoot length, root length, fresh shoot weight, and fresh root weight of rice (Table S2 in S1 File, Figs. S3, S4 in S1 File). BTCP01 exhibited a negative Gram reaction, while BTDR03 tested positive (Table 1). Both strains tested positive for catalase and oxidase tests (Table 1).

Phylogenetic analysis based on the constructed tree using 16S rRNA sequences identified the selected strains as members of the genera *Alcaligenes* and *Metabacillus* (Table 1). A BLASTN search at the GenBank database of NCBI revealed that the sequence of BTCP01, deposited under accession number MW165536, exhibited 99% sequence homology with *Alcaligenes faecalis* (Table 2). The sequences of the isolated strain BTDR03, submitted to GenBank under accession numbers MZ798368, displayed 99% similarity with *Metabacillus indicus* (Fig. S5 in S1 File).

## 3.2. Characterization for plant growth promoting traits of the isolated bacteria

Among the 45 isolates, 20 demonstrated the production of indole-3-acetic acid (IAA) in the presence of L-tryptophan, with concentrations ranging from 13 to 52.78 μg/ml. BTCP01 and

**Table 1. Biochemical and molecular characterization of rice probiotic bacteria isolated from different sources.**

| Strain | Source of isolation | Biochemical analysis | | | | Molecular analysis | | |
|---|---|---|---|---|---|---|---|---|
| | | KOH test | Gram reaction | Catalase test | Oxidase test | Accession No. | Closest strain from gene bank | Sequence similarity (%) |
| **BTCP01** | Surface sterilized shoot of Goose grass (*Elusine indica*) | + | - | ++ | + | MW165536 | *Alcaligenes faecalis* | 99% |
| **BTDR03** | Surface sterilized shoot of Bermuda grass (*Cynodon dactylon*) | - | + | ++ | + | MZ798368 | *Metabacillus indicus* | 99% |

'+' indicates positive response; '−' indicates negative response.

**Table 2. Plant growth-promoting traits of rice probiotic bacteria (Mean ± SE, _n_ = 3).**

| Isolate | Phosphate solubilization (PSI in agar assay) | Potassium solubilization (KSI in agar assay) | IAA production (μg/mL) |
|---|---|---|---|
| BTCP01 | 2.25 ± 0.057a | - | 42.51 ± 0.1a |
| BTDR03 | - | 3.0± 0.577a | 40.86 ± 0.1b |

PSI, Phosphate solubilization index; KSI, Potassium solubilization index; IAA, Indole-3-acetic acid. PSI = (Halo zone + Colony diameter) / Colony diameter, KSI = (Halo zone + Colony diameter) / Colony diameter.

BTDR03 displayed IAA production at levels of 42.51 μg/mL and 40.86 μg/mL, respectively (Table 2, Fig. S4C in S1 File). From this set of isolates, only six exhibited a halo zone on NBRIP agar medium, signifying their phosphate-solubilizing ability. Notably, BTCP01 demonstrated the highest phosphate-solubilizing activity, yielding a PSI value of 2.258 (Table 2, Fig. S4A in S1 File).

Furthermore, among the 20 isolates, only five displayed a halo zone on modified Aleksandrov media (Hu et al., 2006) containing insoluble potassium minerals. Among these, BTDR03 exhibited the highest potassium-solubilizing index (KSI) with a value of 3.0 (Table 2, Fig. S4B in S1 File). However, colony characteristics and seedling assay showed that BTCP01 and BTDR03 were the most prominent strains in terms of their germination, shoot length, root length, shoot fresh weight and dry weight (Table S2 in S1 File). Based on IAA production KSI and seedling assay, we selected BTCP01 and BTDR03 isolates for further investigation.

### 3.3. Genetic identity of probiotic bacteria for growth promotion

Both bacterial isolates (BTCP01 and BTDR03), which exhibited varying levels of growth promotion activities such as IAA production, P and K solubilization, and N-fixation, underwent further scrutiny for the presence of key genes regulating these processes. The isolates were subjected to PCR amplification using gene-specific primers to assess the presence of _nifH_ (responsible for N-fixation) and _ipdC_ (IAA production) genes. Both BTCP01 and BTDR03 isolates were found to harbor the _nifH_ and _ipdC_ genes in their genomes (Table 3). These findings suggest that the majority of the isolates produced IAA, partly through the utilization of the indole-3-pyruvic acid (IPyA) pathway.

### 3.4. Promotion of growth and yield of rice cv. CV. BRRI dhan29

The application of BTCP01 and BTDR03 significantly enhanced the growth and yield of rice (Figs 1 and 2). The tallest plants were observed when 100% of the recommended chemical fertilizer dose was applied to the plants treated with BTDR03 (116 cm), followed closely by BTCP01 (115.33 cm), surpassing the height of uninoculated plants (109.33 cm) under the same fertilizer dose (Fig 3A).

A noteworthy improvement was observed in various growth parameters, including total tiller number per hill, effective tiller number per hill, number of spikelet per panicle, number of filled spikelet per hill, 1000 grain weight, grain yield (t/ha) per pot, shoot fresh and dry weight, and root fresh and dry weight, in bacterial-inoculated plants (Figs 3B–3F and 4A–4F).

**Table 3. Presence (+) or absence (-) of _nifH_ and _ipdC_, genes in bacterial genomes.**

| Isolates | _ipdC_ gene | _nifH_ gene |
|---|---|---|
| BTCP01 | + | + |
| BTDR03 | + | + |

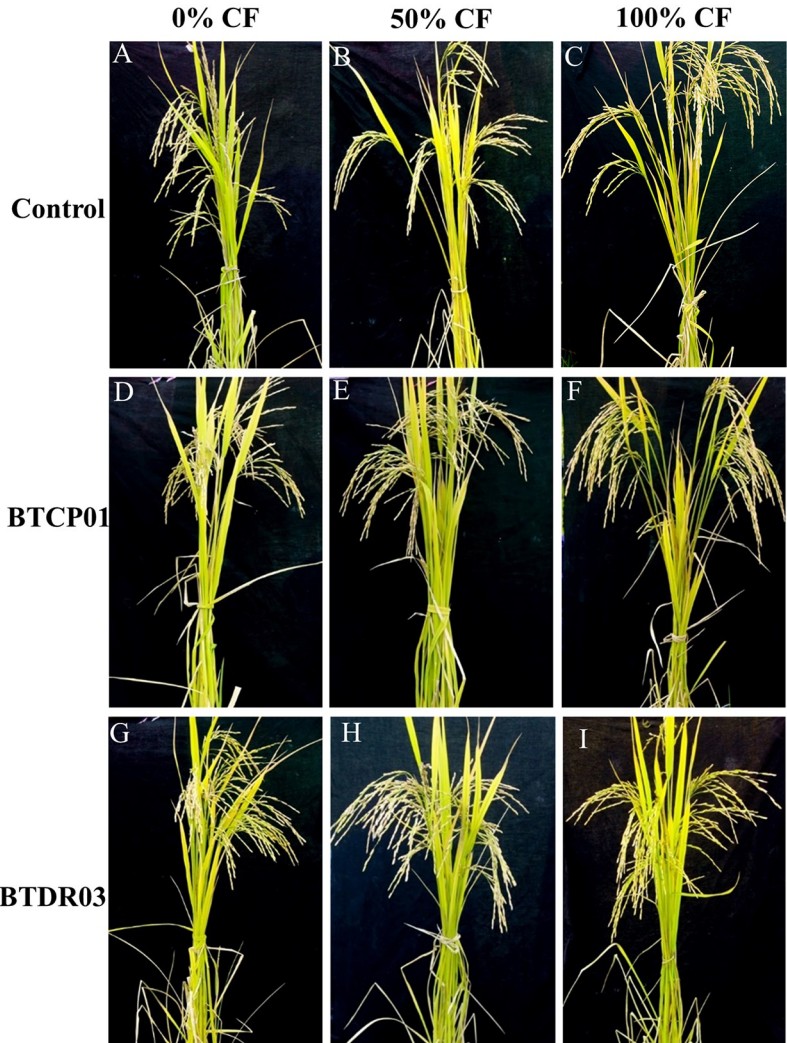

**Fig 1. Effects of BTCP01 and BTDR03 on growth performances of CV.** BRRI dhan29 with 0% (A, D, C), 50% (D, E, F) and 100% (G, H, I) of the recommended doses of chemical fertilizers respectively. *CF (recommended doses of chemical (N, P, K) fertilizers).

Application of 100% of the recommended chemical fertilizer doses to BTCP01 and BTDR03-treated plants significantly increased total tillers per hill and effective tillers per hill compared to untreated controls receiving an equal fertilizer dose (Fig 3C and 3D). Notably, the treatment with 50% of the recommended fertilizer dose along with BTCP01 resulted in more than a 20.68% increase in total tillers per hill (11.66) and a 23.49% increase in effective tillers per hill (10.33). Similar increases were observed for BTDR03, with more than 24% in total tillers per plant (12) and around 27% in effective tillers per plant (10.667) compared to uninoculated plants receiving 100% of the recommended chemical fertilizer dose (Fig 3B and 3C).

The highest number of spikelet per panicle was achieved when 100% of the recommended chemical fertilizer dose was applied to BTDR03-treated plants, followed by BTCP01, with both slightly surpassing the uninoculated control grown with the same fertilizer dose (Fig 3D). Applying 100% of the recommended chemical fertilizer dose to BTDR03-treated plants

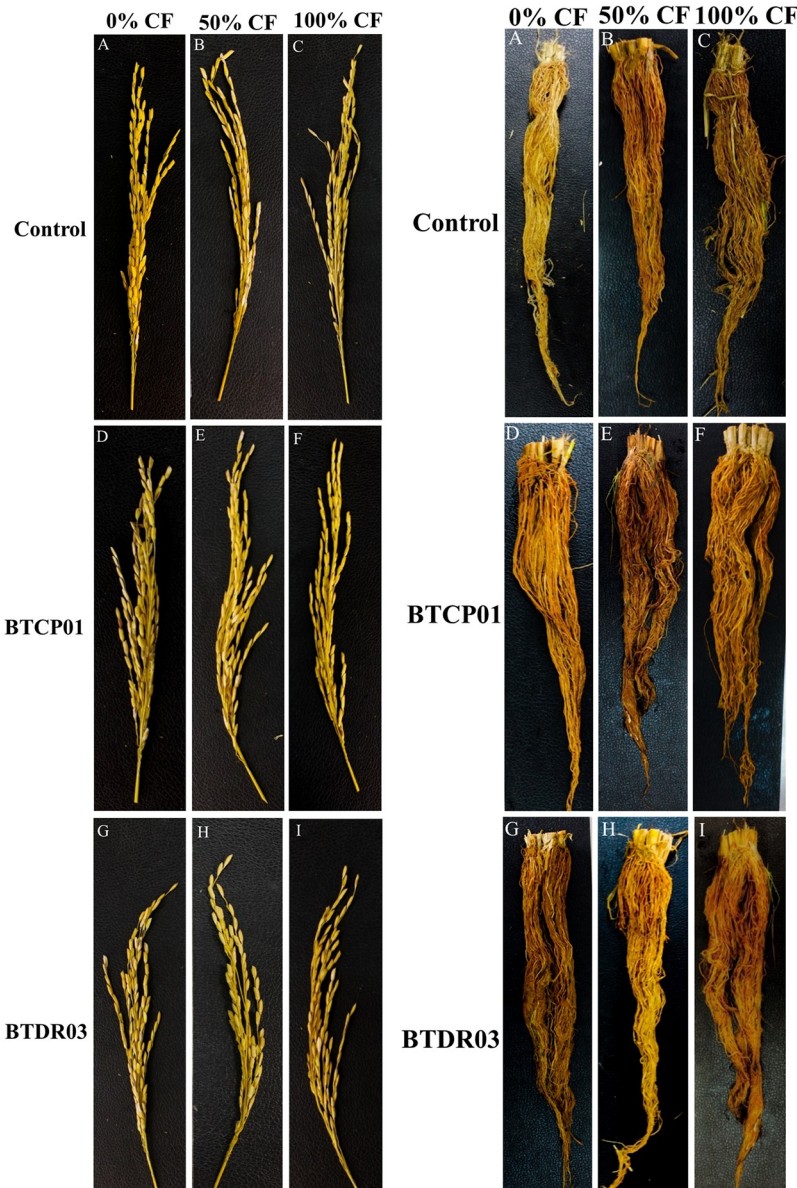

**Fig 2. Effects of BTCP01 and BTDR03 on panicle length and root growth performance of CV.** BRRI dhan29 with 0% (A, D, C), 50% (D, E, F) and 100% (G, H, I) of the recommended doses of chemical fertilizers respectively. *CF (recommended doses of chemical (N, P, K) fertilizers).

produced the highest number of filled spikelet (1089), followed by BTCP01 (1082), significantly surpassing the uninoculated control (970) grown with the same fertilizer dose (Fig 3E). Notably, around 8.31% and 9.78% increases in filled spikelet were observed in BTCP01 and BTDR03-treated plants with 50% of the recommended fertilizer dose, respectively, compared to the untreated control receiving 100% of the recommended fertilizer dose (Fig 3E). Moreover, BTCP01 and BTDR03-treated plants with 50% of the recommended fertilizer dose exhibited a notable 5.64% and 6.75% increase in rice grain yield per pot, respectively, compared to the control treatment (Fig 4B). A similar increasing trend, although not statistically significant,

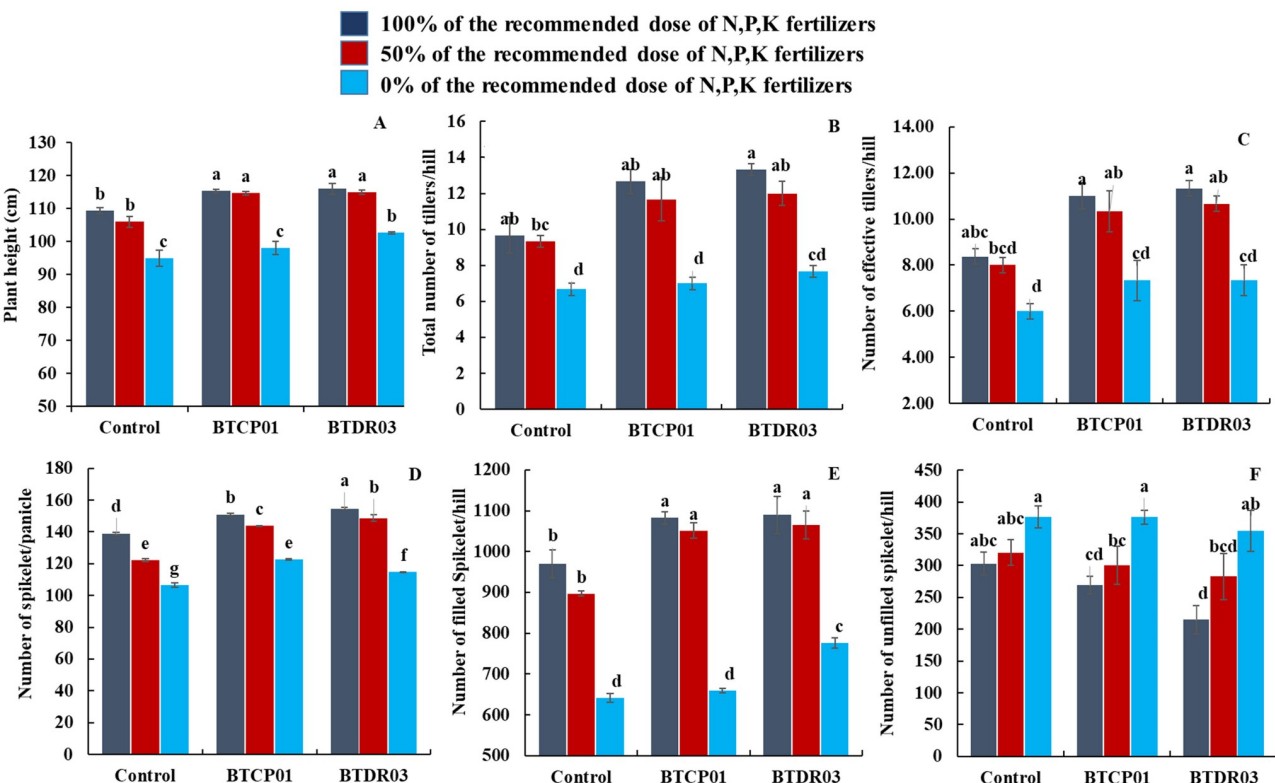

**Fig 3. Effects of BTCP01 and BTDR03 along with different fertilizer doses on various growth parameters of CV. BRRI dhan29.** (A) Effects of probiotic bacteria on plant height of rice, (B) Effects of probiotic bacteria on total number of tillers per hill of rice, (C) Effects of probiotic bacteria on number of effective tillers per hill of rice, (D) Effects of probiotic bacteria on number of spikelet per panicle of rice, (E) Effects of probiotic bacteria on number of filled spikelet per hill of rice, (F) Effects of probiotic bacteria on number of unfilled spikelet per hill of rice. Values (Mean ± SE, $n$ = 3) followed by the same letter(s) in the same graph did not differ significantly at the 0.05 level by the LSD test. Values (Mean ± SE, $n$ = 3) followed by the same letter(s) in the same graph did not differ significantly at the 0.05 level by the LSD test.

was observed in BTCP01-treated plants, while a significant increase was noted in BTDR03-treated plants with 0% of the recommended fertilizer dose, suggesting the potential to reduce major fertilizer use in rice production by up to 50% (Fig 4B).

In addition to grain yield, shoot fresh and dry weight of rice substantially increased in BTCP01-treated plants under 100% of the recommended fertilizer dose, surpassing the untreated control grown under the same conditions (Fig 4C and 4D). Similar trends were observed in treatments with BTCP01 using 100% of the recommended fertilizer dose, significantly enhancing root fresh and dry weight compared to uninoculated control plants grown with similar fertilizer doses (Fig 4E and 4F).

## 4. Discussion

In this study, we isolated 45 endophytic bacteria from native rice weeds and identified two promising rice growth-promoting bacteria, *Alcaligenes faecalis* (BTCP01) and *Metabacillus indicus* (BTDR03), through 16S rRNA gene sequencing. These diazotrophic bacteria were found to significantly enhance seed germination, seedling growth, and ultimately the yield of rice, even with a 50% reduction in N, P, and K fertilizers. We established that the growth-promoting effects were associated with nitrogen fixation (N-fixation), indole-3-acetic acid (IAA) production, and the solubilization of mineral phosphates and potash, suggesting key elements

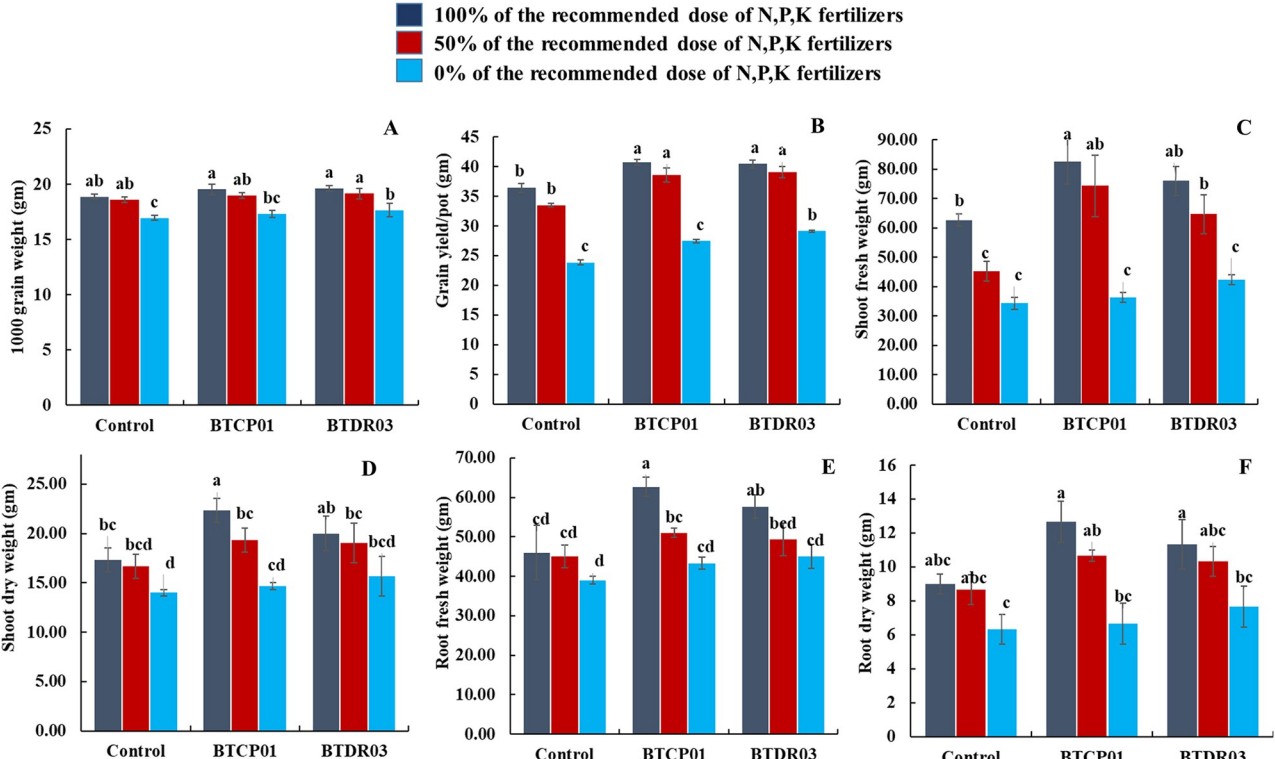

**Fig 4. Effects of BTCP01 and BTDR03 along with different fertilizer doses on various growth parameters of CV. BRRI dhan29.** (A) Effects of probiotic bacteria on 1000 grain weight (g) of rice, (B) Effects of probiotic bacteria on grain yield per pot of rice, (C) Effects of probiotic bacteria on shoot fresh weight (g) of rice, (D) Effects of probiotic bacteria on shoot dry weight (g) of rice, (E) Effects of probiotic bacteria on root fresh weight (g) of rice, (F) Effects of probiotic bacteria on root dry weight of rice. Values (Mean ± SE, $n$ = 3) followed by the same letter(s) in the same graph did not differ significantly at the 0.05 level by the DMRT test using LSD parameter.

contributing to plant growth and productivity [47]. While numerous growth-promoting endophytic bacteria have been previously isolated from various plant sources [8, 9, 47], few have displayed harmful impacts on seed germination and growth. This study identified weed endophytes that significantly improve rice growth and yield, showcasing their potential to reduce fertilizer use by up to 50%, without compromising yield.

A notable discovery in our study was the isolation of diazotrophic *A. faecalis* bacterium (BTCP01) from Goose grass (*Eleusine indica*), a native rice weed, which remarkably increased rice growth and yield with a 50% reduction in major chemical fertilizers (Table 1 and Figs 1–4). *A. faecalis* (BTCP01), initially discovered in feces, has been isolated from various environments, demonstrating its potential as a plant growth-promoting bacteria (PGPB) [48–54]. Our study revealed that BTCP01 enhances rice growth and yield by facilitating nitrogen fixation (as indicated by the presence of the *nifH* gene) and producing indole-3-acetic acid (IAA, detected via the *ipdC* gene). Additionally, *A. faecalis* has been previously identified for its ability to solubilize phosphorus in strains isolated from *Nicotinia glutata*, underscoring its potential as a versatile plant growth-promoting bacterium (PGPB) [55]. Furthermore, it has demonstrated effectiveness as a halotolerant PGPB, supporting the growth of rice, wheat, and canola plants under saline conditions [56–58].

One of the interesting findings of our study was that the application of BTCP01 and BTDR03 with reduced fertilizer doses resulted in statistically equal or higher root length, total tillers per hill, effective tillers per hill, and grain yield compared to untreated control plants

receiving 100% of the recommended doses (Figs 3 and 4). These findings suggest that these two weed endophytic bacteria could effectively reduce N-P-K fertilizer use by up to 50% without compromising rice growth and yield. Coinoculation with various strains of *Burkholderia* spp. and *Pseudomonas aeruginosa* from different weeds has been reported to enhance plant growth and yield in several crops, emphasizing the potential for field evaluations [59–62]. However, we for the first time demonstrated that weed endophytic bacteria *A*. *faecalis* and *M*. *indicus* isolated from the rice weeds have potential for reduction in chemical fertilizers in rice. A further field level evaluation of these two bacteria either alone or in combination are needed to confirm the potentials as candidates for biofertilization in rice.

To see the mechanistic insights of the higher yield in rice by the weed endophytic bacteria, we checked whether they possess any genetic traits in their genome association with plant growth promotion. Interestingly, we detected *nifH* and *ipdC* genes in the genomes of BTCP01 and BTDR03 that suggests their potential to fix nitrogen and produce phytohormone IAA. While the growth and grain yield enhancement of treated plants may be associated with increased nutrient uptake [63], additional mechanistic studies are needed to uncover the full potential of these bacterial isolates as bioinoculants.

Notably, some weed endophytes in our study severely suppressed rice seed germination (Table S2 in S1 File). Given that weeds are known competitors of rice and can inhibit seed germination and crop growth, understanding the molecular basis of weed endophytes inhibiting rice seed germination warrants further investigation. Although allelopathic effects of weeds through phytotoxic secondary metabolites have been reported [64], reports on allelopathic effects of weed endophytes are limited. Phytotoxic compounds from weed endophytes may offer new herbicide candidates. Our findings underscore the importance of discovering novel weed endophytes as valuable bioresources for sustainable agriculture, contributing to a reduction in the use of synthetic agrochemicals that pose threats to soil and environmental health.

In conclusion, this study represents the first identification and characterization of two weed endophytic bacteria, *A*. *faecalis* and *M*. *indicus*, with the ability to enhance the growth and yield of rice under 50% reduced doses of N, P, and K chemical fertilizers. The underlying mechanisms of their beneficial effects encompass IAA production, atmospheric N-fixation, and mineral P and K solubilization. Additionally, the presence of *nifH* and *ipdC* genes correlates with their growth-promoting activities on rice. Consequently, these endophytic bacteria, sourced from rice-associated weeds, hold significant promise for reducing chemical fertilizer usage in sustainable rice production. Enhancing grain yield while reducing chemical fertilization through the use of plant growth-promoting bacteria (PGPB) strains, notably *A*. *faecalis* (BTCP01) and *M*. *indicus* (BTDR03), holds promise for advancing sustainable agricultural practices. However, before recommending these strains as biofertilizers for rice cultivation, a comprehensive, multi-location field evaluation is essential. Additionally, investigating the synergistic effects of co-inoculating these two weed endophytes on rice represents a compelling avenue for further research.

## Supporting information

**S1 File.**
(DOCX)

## Acknowledgments

The authors are thankful to the Bangladesh Academy of Sciences for funding this work through a BAS-USDA-PAL project no. CR-11.

## Author Contributions

**Conceptualization:** Tofazzal Islam.

**Data curation:** Nur Uddin Mahmud, Md. Nurealam Siddiqui.

**Formal analysis:** Kaniz Fatema, Nur Uddin Mahmud, Tahsin Islam Sakif.

**Funding acquisition:** Tofazzal Islam.

**Investigation:** Kaniz Fatema, Nur Uddin Mahmud.

**Methodology:** Kaniz Fatema, Aniruddha Sarker.

**Project administration:** Tofazzal Islam.

**Resources:** Andrew G. Sharpe, Tofazzal Islam.

**Software:** Md. Nurealam Siddiqui, Tahsin Islam Sakif.

**Supervision:** Tofazzal Islam.

**Writing – original draft:** Kaniz Fatema.

**Writing – review & editing:** Dipali Rani Gupta, Md. Nurealam Siddiqui, Tahsin Islam Sakif, Aniruddha Sarker, Andrew G. Sharpe, Tofazzal Islam.

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
