## [Decision Letter · Decision Letter 0]

8 Jan 2024

PONE-D-23-42245Enhancing Rice Growth and Yield with Weed Endophytic Bacteria Alcaligenes faecalis and Metabacillus indicus Under Reduced Chemical FertilizationPLOS ONE

Dear Dr. Islam,

Thank you for submitting your manuscript to PLOS ONE. After careful consideration, we feel that it has merit but does not fully meet PLOS ONE’s publication criteria as it currently stands. Therefore, we invite you to submit a revised version of the manuscript that addresses the points raised during the review process. Please submit your revised manuscript by Feb 22 2024 11:59PM. If you will need more time than this to complete your revisions, please reply to this message or contact the journal office at plosone@plos.org. Please include the following items when submitting your revised manuscript:A rebuttal letter that responds to each point raised by the academic editor and reviewer(s). You should upload this letter as a separate file labeled 'Response to Reviewers'.A marked-up copy of your manuscript that highlights changes made to the original version. You should upload this as a separate file labeled 'Revised Manuscript with Track Changes'.An unmarked version of your revised paper without tracked changes. You should upload this as a separate file labeled 'Manuscript'.

We look forward to receiving your revised manuscript.

Kind regards,

Sofia Isabel Almeida Pereira

Academic Editor

PLOS ONE

Journal Requirements:

3. Thank you for stating the following in the Acknowledgments Section of your manuscript: "The authors are thankful to Bangladesh Academy of Sciences (BAS) for the funding of this work under a project No. BAS-USDA PALS CR-11.".

Please remove any funding-related text from the manuscript and let us know how you would like to update your Funding Statement. Currently, your Funding Statement reads as follows: "The authors received no specific funding for this work."

Reviewers' comments:

Reviewer's Responses to Questions

**Comments to the Author**

1. Is the manuscript technically sound, and do the data support the conclusions?

Reviewer #1: Yes

Reviewer #2: Partly

2. Has the statistical analysis been performed appropriately and rigorously? 

Reviewer #1: Yes

Reviewer #2: I Don't Know

3. Have the authors made all data underlying the findings in their manuscript fully available?

Reviewer #1: Yes

Reviewer #2: Yes

4. Is the manuscript presented in an intelligible fashion and written in standard English?

Reviewer #1: Yes

Reviewer #2: No

5. Review Comments to the Author

Reviewer #1: 1. Clarity of Experimental Design:

- The experimental design and methodology need further clarification. Provide a detailed description of the isolation process for endophytic bacteria from native weeds, including specific procedures, media composition, and isolation conditions. Additionally, elaborate on the criteria used for selecting the two promising strains (BTCP01 and BTDR03) and the rationale behind their identification.

2. Genetic Characterization:

- Strengthen the genetic characterization section by providing more details on the detection and analysis of the nifH and ipdC genes in the genomes of BTCP01 and BTDR03. Include the primers used, amplification conditions, and the significance of these genes in the context of plant growth promotion.

3. Data Presentation:

- Revise the presentation of results to enhance clarity. Provide clear tables or figures summarizing the mineral phosphate and potash solubilization activities of the isolated strains. Include statistical analyses where applicable and specify the significance levels.

4. Root Dipping and Spraying Experiment:

- Elaborate on the methodology employed for the application of isolates BTCP01 and BTDR03 through root dipping and spraying at the flowering stage. Include details on the concentration, frequency, and duration of the applications.

5. Agronomic Performance Assessment:

- Provide a more comprehensive analysis of the agronomic performance of rice variety BRRI dhan29. Include additional parameters such as plant height, tiller number, and any observed morphological changes. Consider incorporating statistical analyses to support the significance of the observed effects.

6. Comparison with Control:

- Clearly outline the control conditions in the experiment and provide a detailed comparison with the treated plants. Specify the outcomes in terms of growth parameters and grain yields.

7. Discussion on Sustainable Agriculture:

- Expand the discussion on the potential implications of using weed endophytic bacteria as biofertilizers for sustainable agriculture. Address the broader context of reducing chemical fertilizer dependency and its impact on environmental sustainability.

8. Concluding Remarks:

- Strengthen the concluding remarks by summarizing the key findings and their implications for sustainable rice production. Discuss the significance of the observed increase in grain yields and its potential contribution to agricultural practices.

9. Language and Style:

- Review and improve the clarity of language and style throughout the manuscript. Ensure consistency in terminology and provide definitions or explanations for any specialized terms. Proofread for grammatical accuracy and overall readability.

Addressing these major modifications will enhance the clarity, rigor, and overall quality of the manuscript.

Reviewer #2: -It would have been better to design the experiment using a treatment containing a mixture of the two isolates

-It was necessary to ensure that the surface sterilization process of plant samples was successful; otherwise you cannot confirm that the isolated bacteria are endophytes.

-Lines 99, 100: you reported that “seeds of the rice variety CV. BRRI dhan29 were procured from the Bangladesh Rice Research Institute (BRRI) for use in pot experiments to isolate endophytic bacteria” indicating isolation from rice seeds, while you reported isolation from grasses. Explain?

-In the indole production experiment, why did you use that concentration of L-tryptophan (1 ml of 0.2%)?

-The results should include a summary of the seedling assay results, especially since you chose the best isolates according to the results of the germination rate

- Line 107, 108, 238: If the nutrient medium contains agar, “nutrient broth agar medium” must be

Corrected to “nutrient agar medium”

-Line 270: “CPR01 and DRB03”, these codes do not match any of the codes for the 45 isolates included in the study?

- The results should be presented in a simple and direct way that highlights the important points of the manuscript

- The discussion must be improved

Typo errors

Line 33: write “Eleusine indica” in italic

Line 96: correct “hexanda” to “hexandra”, correct “Burnyard” to “Barnyard”, correct “crussgalli” to “crus-galli”

Line 184: delete “[PSI”

Line 197: delete “[KSI”

Line 623: correct rice variety name “BRRI dhan29”

Line 329: correct “Nicotinia” to “Nicotiana”

The name of the rice variety is written as “BRRI dhan29 “, CV. BRRI dhan29” and sometimes as “cv. BRRI dhan29”, check it

Fig. 1, 2: letters (A-I) in the figure title corresponding to percentage of the chemical fertilizer does not agree with the meaning of the letters on the figures

Fig. 3 B; delete the duplicated letter (B)

Line 421: correct “nbacteria” to “bacteria”

Lines 553, 559: write “Bacillus” in italic

Line 575: write “Agrobacterium” in italic

Correct title of Fig. S3 “Effects of various bacterial isolates on the rice seedlings”

-The title written in the supplementary material is different from the title written in the main manuscript

References

-References No. 5, 7, 8, 9, 34, 47, 55, 56, 61, 62, 63 and 64 are not present in the text.

-References cited in the text and missed from the list;

-Line 131: “Bergey et al. (1994).”

-Line 139: “Maniatis et al. (1982).”

-Line 143: “(Reysenbach et al., 1992)”

-Line 153: “(Saitou and Nei, 1987), (Kumar et al., 2016)”

-Line 168: Bric et al., 1991

6. PLOS authors have the option to publish the peer review history of their article (what does this mean?). If published, this will include your full peer review and any attached files.

Reviewer #1: **Yes: **Arun Karnwal

Reviewer #2: **Yes: **Ahmed Mohamed Eid

---

## [Author Response · Author response to Decision Letter 0]

15 Feb 2024

Dated: 9 February 2024

Editorial Office

PLOS ONE

Subject: Revision of manuscript # PONE-D-23-42245

Dear Dr Sofia Isabel Almeida Pereira,

First of all, we gratefully appreciate your time and interest on handling our manuscript. We have revised the above-mentioned manuscript according to the suggestions of the two reviewers. A point-wise reply to the reviewer´s comments is presented at the end of this letter. We edited the manuscript text and re-arranged the Figure in order to provide an improved version that will hopefully meet the standards for publication at PLOS ONE. All alterations were made highlighting by the red color in Word, and are therefore properly highlighted in the revised manuscript.

All listed co-authors have approved the resubmission of this manuscript. The authors declare no conflict of interest.

We gratefully appreciate your time and interest. We look forward to hearing from you.

Sincerely,

Tofazzal Islam

Fellow of the American Phytopathological Society (APS), Bangladesh Academy of Sciences (BAS), The World Academy of Sciences (TWAS), and Bangladesh Academy of Agriculture (BAAG)

&

Professor and Founding Director, Institute of Biotechnology and Genetic Engineering (IBGE),

Bangabandhu Sheikh Mujibur Rahman Agricultural University, Gazipur-1706

BANGLADESH

Tel. +88-02-9205310-14 Extn. 2252

Fax: +88-02-9205333

Cell: +88-0171-4001414

Editor: Physiologia Plantarum

Associate Editor: Frontiers in Microbiology

Editor: Scientific Reports

Academic Editor: PLOS ONE

https://ibge.bsmrau.edu.bd/tofazzal/

Our Response to reviewer comments

Reviewer #1: 

1. Clarity of Experimental Design:

- The experimental design and methodology need further clarification. Provide a detailed description of the isolation process for endophytic bacteria from native weeds, including specific procedures, media composition, and isolation conditions. Additionally, elaborate on the criteria used for selecting the two promising strains (BTCP01 and BTDR03) and the rationale behind their identification.

Response: Thank you for your nice comments. We have provided a detailed description of the isolation process for endophytic bacteria from native weeds, including specific procedures, media composition, and isolation conditions under the headline 2.2. Collection of plant materials and isolation of bacteria (Line 90-109). However, the criteria used for selecting the two promising strains (BTCP01 and BTDR03) and the rationale behind their identification were elaborated under the headline 3.1. Isolation, biochemical and molecular characterization of weed endophytic bacteria (Line 241-257).

2. Genetic Characterization:

- Strengthen the genetic characterization section by providing more details on the detection and analysis of the nifH and ipdC genes in the genomes of BTCP01 and BTDR03. Include the primers used, amplification conditions, and the significance of these genes in the context of plant growth promotion.

Response: We have provided the detailed on the detection and analysis of nifH and ipdC genes in the genomes of BTCP01 and BTDR03 in Supplementary Table S1. The significance of these genes in the context of plant growth promotion were explained in Line 331-336.

3. Data Presentation:

- Revise the presentation of results to enhance clarity. Provide clear tables or figures summarizing the mineral phosphate and potash solubilization activities of the isolated strains. Include statistical analyses where applicable and specify the significance levels.

Response: Thank you. We have provided the detailed on clear tables and figures summarizing the mineral phosphate and potash solubilization activities of the isolated strains in Supplementary Table 2, where we have also included statistical analysis for identifying the significance levels by different lettering. 

4. Root Dipping and Spraying Experiment:

- Elaborate on the methodology employed for the application of isolates BTCP01 and BTDR03 through root dipping and spraying at the flowering stage. Include details on the concentration, frequency, and duration of the applications.

Response: Thank you. We have provided the detailed in the Line number 215-226.

5. Agronomic Performance Assessment:

- Provide a more comprehensive analysis of the agronomic performance of rice variety BRRI dhan29. Include additional parameters such as plant height, tiller number, and any observed morphological changes. Consider incorporating statistical analyses to support the significance of the observed effects.

Response: We have provided the detail agronomic performances under the headline 3.4. Promotion of growth and yield of rice cv. BRRI dhan29 (Line280-314).

6. Comparison with Control:

- Clearly outline the control conditions in the experiment and provide a detailed comparison with the treated plants. Specify the outcomes in terms of growth parameters and grain yields.

Response: Thank you for your crucial comments. We have provided the detailed under the headline 3.4. Promotion of growth and yield of rice cv. BRRI dhan29 (Line 280-314).

7. Discussion on Sustainable Agriculture:

- Expand the discussion on the potential implications of using weed endophytic bacteria as biofertilizers for sustainable agriculture. Address the broader context of reducing chemical fertilizer dependency and its impact on environmental sustainability.

Response:

Thank you for raising this important issue. We have added the context of reducing fertilizer dependency using plant-growth promoting bacteria in introduction (Line 60-66). To avoid the redundancy, we did not discuss this issue again in discussion section.

8. Concluding Remarks:

- Strengthen the concluding remarks by summarizing the key findings and their implications for sustainable rice production. Discuss the significance of the observed increase in grain yields and its potential contribution to agricultural practices.

Response: We have now added a key finding in concluding remarks in Line (369-375).

9. Language and Style:

- Review and improve the clarity of language and style throughout the manuscript. Ensure consistency in terminology and provide definitions or explanations for any specialized terms. Proofread for grammatical accuracy and overall readability. Addressing these major modifications will enhance the clarity, rigor, and overall quality of the manuscript.

Response: Thank you. We have improved the clarity of language and style throughout the manuscript thus definitely enhanced the clarity, rigor, and overall quality of the manuscript.

#############################################################################

Reviewer #2: 

-It would have been better to design the experiment using a treatment containing a mixture of the two isolates.

Response: Thank you for your valuable comments. For a clear understanding, we provided the isolate performances under different fertilizer levels in Figures 3 and 4. 

-It was necessary to ensure that the surface sterilization process of plant samples was successful; otherwise, you cannot confirm that the isolated bacteria are endophytes.

Response: Thank you for your valuable comments. Yes, surface sterilization was successful, therefore we found significant differences in germination rate, shoot length, root length, shoot fresh weight, and root fresh weight (Supplementary Table S2).

-Lines 99, 100: you reported that “seeds of the rice variety CV. BRRI dhan29 were procured from the Bangladesh Rice Research Institute (BRRI) for use in pot experiments to isolate endophytic bacteria” indicating isolation from rice seeds, while you reported isolation from grasses. Explain?

Response: Sorry for this mistake. Now we have corrected the statement (Line 99-100).

-In the indole production experiment, why did you use that concentration of L-tryptophan (1 ml of 0.2%)?

Response: Thank you for your valuable comments. The production of indole-3-acetic acid (IAA), a plant hormone, by bacteria using L-tryptophan as a precursor. According to some studies, the optimal conditions for IAA production vary depending on the bacterial strain and the culture medium. For example, Rhizobium sp. produced the highest amount of IAA (90.21 μg/ml) in culture media supplemented with 0.2% L-tryptophan and at an initial pH of 9.

-The results should include a summary of the seedling assay results, especially since you chose the best isolates according to the results of the germination rate.

Response: Thank you for your valuable comments. We have added a summary in Line 269-273.

- Line 107, 108, 238: If the nutrient medium contains agar, “nutrient broth agar medium” must be

Corrected to “nutrient agar medium”

Response: Thank you. We have corrected the error throughout the manuscript. 

-Line 270: “CPR01 and DRB03”, these codes do not match any of the codes for the 45 isolates included in the study?

Response: Thank you. We have corrected the error.

- The results should be presented in a simple and direct way that highlights the important points of the manuscript

- The discussion must be improved

Typo errors

Line 33: write “Eleusine indica” in italic

Response: Thank you. We have corrected the error. 

Line 96: correct “hexanda” to “hexandra”, correct “Burnyard” to “Barnyard”, correct “crussgalli” to “crus-galli”

Response: Thank you. We have corrected the error. 

Line 184: delete “[PSI”

Response: Thank you. We have corrected the error. 

Line 197: delete “[KSI”

Response: Thank you. We have corrected the error. 

Line 623: correct rice variety name “BRRI dhan29”

Response: Thank you. We have corrected the error.

Line 329: correct “Nicotinia” to “Nicotiana”

Response: Thank you. We have corrected the error.

-The name of the rice variety is written as “BRRI dhan29 “, CV. BRRI dhan29” and sometimes as “cv. BRRI dhan29”, check it

Response: Thank you. We have corrected the error.

-Fig. 1, 2: letters (A-I) in the figure title corresponding to percentage of the chemical fertilizer does not agree with the meaning of the letters on the figures

Response: Thank you. We kept lettering for quick understanding and matching with figure legend.

-Fig. 3 B; delete the duplicated letter (B)

Response: Deleted.

Line 421: correct “nbacteria” to “bacteria”

Response: Thank you. We have corrected the error.

Lines 553, 559: write “Bacillus” in italic

Response: Thank you. We have corrected the error.

Line 575: write “Agrobacterium” in italic

Response: Thank you. We have corrected the error.

-Correct title of Fig. S3 “Effects of various bacterial isolates on the rice seedlings”

Response: Thank you. Corrected.

-The title written in the supplementary material is different from the title written in the main manuscript

Response: Thank you for pointing out this mistake. Now it has been corrected.

References

-References No. 5, 7, 8, 9, 34, 47, 55, 56, 61, 62, 63 and 64 are not present in the text.

-References cited in the text and missed from the list;

-Line 131: “Bergey et al. (1994).”

-Line 139: “Maniatis et al. (1982).”

-Line 143: “(Reysenbach et al., 1992)”

-Line 153: “(Saitou and Nei, 1987), (Kumar et al., 2016)”

-Line 168: Bric et al., 1991

Response: Reference list matching with text has corrected.

##############################################################################

---

## [Decision Letter · Decision Letter 1]

6 Mar 2024

PONE-D-23-42245R1Enhancing Rice Growth and Yield with Weed Endophytic Bacteria Alcaligenes faecalis and Metabacillus indicus Under Reduced Chemical FertilizationPLOS ONE

Dear Dr. Islam,

Thank you for submitting your manuscript to PLOS ONE. After careful consideration, we feel that it has merit but does not fully meet PLOS ONE’s publication criteria as it currently stands. Therefore, we invite you to submit a revised version of the manuscript that addresses the points raised during the review process.

We look forward to receiving your revised manuscript.

Kind regards,

Sofia Isabel Almeida Pereira

Academic Editor

PLOS ONE

Reviewers' comments:

Reviewer's Responses to Questions

**Comments to the Author**

1. If the authors have adequately addressed your comments raised in a previous round of review and you feel that this manuscript is now acceptable for publication, you may indicate that here to bypass the “Comments to the Author” section, enter your conflict of interest statement in the “Confidential to Editor” section, and submit your "Accept" recommendation.

Reviewer #1: All comments have been addressed

2. Is the manuscript technically sound, and do the data support the conclusions?

Reviewer #1: Yes

3. Has the statistical analysis been performed appropriately and rigorously? 

Reviewer #1: Yes

4. Have the authors made all data underlying the findings in their manuscript fully available?

Reviewer #1: Yes

5. Is the manuscript presented in an intelligible fashion and written in standard English?

Reviewer #1: Yes

6. Review Comments to the Author

Reviewer #1: The study investigating the potential of endophytic bacteria from native weeds as biofertilizers presents valuable insights into sustainable agricultural practices. The manuscript is well-written and addresses an important gap in the literature regarding the plant growth-promoting abilities of weed endophytes. However, to strengthen the impact and clarity of the findings, few minor modifications are recommended:

1. Clarification of Methodology: Provide a more detailed description of the isolation and identification methods used for endophytic bacteria from native weeds. This will enhance reproducibility and aid readers in understanding the experimental procedures.

2. Taxonomic Identification: While the tentative identification of the two promising strains (BTCP01 and BTDR03) based on 16S rRNA gene phylogeny is informative, consider additional validation methods, such as whole-genome sequencing or multilocus sequence analysis, to confirm their taxonomic classification.

3. Quantitative Data Presentation: Present quantitative data on mineral phosphate and potash solubilization activities, as well as IAA production, in a clear and organized manner. Tables or graphs summarizing these results would facilitate interpretation and comparison between isolates.

4. Gene Detection Analysis: Provide more details on the methodology and significance of detecting the nifH and ipdC genes in the genomes of the selected isolates. Discuss the potential role of these genes in plant-microbe interactions and their implications for biofertilizer applications.

5. Agronomic Performance Assessment: Expand on the methods used to evaluate the agronomic performance of rice variety BRRI dhan29 following inoculation with endophytic bacterial strains. Include parameters such as plant height, root morphology, and nutrient uptake efficiency to comprehensively assess the plant growth-promoting effects.

6. Statistical Analysis and Interpretation: Enhance the statistical analysis by including appropriate tests to determine the significance of differences observed in agronomic parameters between treatment groups and control. Provide a detailed interpretation of the results, discussing the practical implications for sustainable rice production.

Addressing these minor modifications will enhance the rigor, clarity, and impact of the study, further advancing our understanding of the potential of weed endophytic bacteria as biofertilizers in sustainable agriculture.

7. PLOS authors have the option to publish the peer review history of their article (what does this mean?). If published, this will include your full peer review and any attached files.

Reviewer #1: **Yes: **Arun Karnwal

---

## [Author Response · Author response to Decision Letter 1]

19 Mar 2024

Responses to the Reviewer’s Comments

Title of the Manuscript: Enhancing Rice Growth and Yield with Weed Endophytic Bacteria Alcaligenes faecalis and Metabacillus indicus Under Reduced Chemical Fertilization

Manuscript No. PONE-D-23-42245R1

Dear Editor,

We are deeply grateful for the opportunity to revise our manuscript for your esteemed journal. We extend our sincere appreciation to you and the diligent reviewers for their invaluable feedback, which has significantly enhanced the quality of our work. We have meticulously addressed each comment and incorporated necessary revisions to improve the clarity and precision of our manuscript.

Please find below a detailed point-by-point response to the reviewers' comments:

Review Comments to the Author

Reviewer #1: The study investigating the potential of endophytic bacteria from native weeds as biofertilizers presents valuable insights into sustainable agricultural practices. The manuscript is well-written and addresses an important gap in the literature regarding the plant growth-promoting abilities of weed endophytes. However, to strengthen the impact and clarity of the findings, few minor modifications are recommended:

Our response: Many thanks for the encouraging comments and valuable comments for improvement of the impact and clarity of the findings.

1. Clarification of Methodology: Provide a more detailed description of the isolation and identification methods used for endophytic bacteria from native weeds. This will enhance reproducibility and aid readers in understanding the experimental procedures.

Our response: Thank you. We have provided a detailed description of the isolation process for endophytic bacteria from native weeds, including specific procedures, media composition, and isolation conditions under the headline 2.2. Collection of plant materials and isolation of bacteria.

2. Taxonomic Identification: While the tentative identification of the two promising strains (BTCP01 and BTDR03) based on 16S rRNA gene phylogeny is informative, consider additional validation methods, such as whole-genome sequencing or multilocus sequence analysis, to confirm their taxonomic classification.

Our response: We value the reviewer's insightful suggestion. While whole-genome sequencing offers more definitive taxonomic classification, its implementation involves considerable time, complexity in data analysis, and costs. Regrettably, these constraints preclude its inclusion within the timeframe allocated for revising this manuscript. Given that 16S rRNA gene sequencing is generally acceptable for preliminary identification of environmental bacteria, we will explore the recommended whole-genome sequencing and data annotation in our future projects.

3. Quantitative Data Presentation: Present quantitative data on mineral phosphate and potash solubilization activities, as well as IAA production, in a clear and organized manner. Tables or graphs summarizing these results would facilitate interpretation and comparison between isolates.

Our response: We have included detailed tables and figures in Supplementary Table 2, summarizing the mineral phosphate and potash solubilization activities of the isolated strains. Statistical analyses were conducted where applicable, and significance levels were specified.

4. Gene Detection Analysis: Provide more details on the methodology and significance of detecting the nifH and ipdC genes in the genomes of the selected isolates. Discuss the potential role of these genes in plant-microbe interactions and their implications for biofertilizer applications.

Our response: Thank you for this important suggestion. We have provided the details on 2.6. Design of primers for amplification of nifH and ipdC genes.

5. Agronomic Performance Assessment: Expand on the methods used to evaluate the agronomic performance of rice variety BRRI dhan29 following inoculation with endophytic bacterial strains. Include parameters such as plant height, root morphology, and nutrient uptake efficiency to comprehensively assess the plant growth-promoting effects.

Our response: This is indeed a good suggestion. However, in this project, we did not evaluate the root morphology and nutrient uptake efficiency. This suggestion will be helpful in our next project.

6. Statistical Analysis and Interpretation: Enhance the statistical analysis by including appropriate tests to determine the significance of differences observed in agronomic parameters between treatment groups and control. Provide a detailed interpretation of the results, discussing the practical implications for sustainable rice production.

Our response: Thank you so much for this suggestion. Our main target was to identify potential bacteria which can improve the growth and yield with various fertilizer doses. And we have provided detailed interpretation under the headline 3.4. Promotion of growth and yield of rice cv. CV. BRRI dhan29.

We hope that our revised version will be accepted by the reviewers and you. Should you have any further queries and suggestions, please feel free to contact me.

Thank you once again for your time and consideration of our manuscript.

Best regards,

Tofazzal Islam March 19, 2024

..........................................

Md Tofazzal Islam, Ph D

Fellow of the American Phytopathological Society (APS), Bangladesh Academy of Sciences (BAS), The World Academy of Sciences (TWAS), and Bangladesh Academy of Agriculture (BAAG)

&

Professor and Founding Director, Institute of Biotechnology and Genetic Engineering (IBGE),

Bangabandhu Sheikh Mujibur Rahman Agricultural University, Gazipur-1706

BANGLADESH

Tel. +88-02-9205310-14 Extn. 2252

Fax: +88-02-9205333

Cell: +88-0171-4001414

Editor: Physiologia Plantarum

Associate Editor: Frontiers in Microbiology

Editor: Scientific Reports

Academic Editor: PLOS ONE

https://ibge.bsmrau.edu.bd/tofazzal/

http://www.researchgate.net/profile/Md_Tofazzal_Islam

http://orcid.org/0000-0002-7613-0261

https://www.apsnet.org/members/give-awards/awards/2022Awardees/Pages/Fellow_Islam_Tofazzal.aspx

https://twas.org/directory/islam-md-tofazzal

---

## [Editor Report · Decision Letter 2]

24 Mar 2024

Enhancing Rice Growth and Yield with Weed Endophytic Bacteria Alcaligenes faecalis and Metabacillus indicus Under Reduced Chemical Fertilization

PONE-D-23-42245R2

Dear Dr.Islam, 

We’re pleased to inform you that your manuscript has been judged scientifically suitable for publication and will be formally accepted for publication once it meets all outstanding technical requirements.

Kind regards,

Sofia Isabel Almeida Pereira

Academic Editor

PLOS ONE

---

## [Editor Report · Acceptance letter]

29 Mar 2024

PONE-D-23-42245R2 

PLOS ONE

Dear Dr. Islam, 

I'm pleased to inform you that your manuscript has been deemed suitable for publication in PLOS ONE. Congratulations! Your manuscript is now being handed over to our production team.

Kind regards, 

on behalf of

Dr. Sofia Isabel Almeida Pereira 

Academic Editor

PLOS ONE